# Knowledge, Attitudes, and Practices among Pharmacy and Non-Pharmacy Interns in Saudi Arabia Regarding Antibiotic Use and Antibiotic Resistance: A Cross-Sectional Descriptive Study

**DOI:** 10.3390/healthcare11091283

**Published:** 2023-04-29

**Authors:** Laila Albalawi, Abeer S. Alhawiti, Dania Alnasser, Jawaher Alhumaidi, Tahani Alrashidi, Awatif M. Alnawmasi, Mostafa A. S. Ali, Saleh Alqifari, Hanan Alshareef

**Affiliations:** 1Faculty of Pharmacy, University of Tabuk, Tabuk 71491, Saudi Arabia; 2Pharmacy Practice Department, Faculty of Pharmacy, University of Tabuk, Tabuk 71491, Saudi Arabia; 3Department of Clinical Pharmacy, Faculty of Pharmacy, Assiut University, Assiut 71526, Egypt

**Keywords:** interns, antibiotics, antibiotic resistance, Saudi Arabia

## Abstract

This cross-sectional descriptive study aims to explore the knowledge, attitudes, and practices among pharmacy, dentistry, medicine, and nursing interns in Saudi Arabia regarding antibiotic use and antibiotic resistance. Interns received a soft copy of a structured and validated self-administered questionnaire using an online survey platform. A total of 266 interns responded to the questionnaire. On average, the participants achieved good scores in the knowledge and practice domains, followed by the attitude domain. The average knowledge percentage was 76.1% (SD 17.1) compared to 84.6% (SD 20.5) for practices and 61.5% (SD 23.2) for attitudes. The results suggest that pharmacy interns had insignificantly better overall scores compared to non-pharmacy interns for knowledge, attitudes, and practices. This study shows that the scores for knowledge, attitudes, and practices of pharmacy, dentistry, medicine, and nursing interns regarding antibiotic use and resistance were high overall. However, interns’ belief in their responsibility in preventing antibiotic resistance is lacking.

## 1. Introduction

Antimicrobials are medications used to treat or prevent infectious diseases caused by microorganisms, including bacteria, viruses, fungi, and parasites [1]. Inappropriate use of antimicrobials results in the emergence of resistance, which is a serious condition that occurs when the microorganisms change over time and no longer respond to antimicrobials, making the infection harder to treat and increasing the risk of spread, the severity, and the mortality of the disease [2]. New resistance mechanisms are evolving and spreading worldwide, making common infectious diseases difficult to treat [2,3]. Antimicrobial resistance is increasing at an alarming rate worldwide, leading to higher hospital expenditures, lengths of stay, and increased healthcare costs for patients and their families [1,3].

At least 2.8 million individuals in the United States become infected with antimicrobial-resistant fungi or bacteria each year, with more than 35,000 people dying as a result [1,4]. According to the Centers for Disease Control and Prevention (CDC), antimicrobial resistance-related costs amount to more than USD 4.6 billion annually in the United States [4]. If the current trend of inappropriate and excessive antibiotic usage continues, it is anticipated that 10 million people will die worldwide by 2050 [5,6].

The scientific understanding of the issues linked to antibiotic use must be improved [7]. The majority of initiatives aiming to control the use of antimicrobials have focused on prescribers [4]. In Saudi Arabia, community pharmacists cannot dispense antibiotics without medical prescriptions; these regulatory policies were developed in response to the occurrence of widespread antimicrobial resistance and to improve antibiotic prescription procedures [7,8]. Antibiotic policies at the national and international levels and educational initiatives all form part of the guidelines, strategies, and instructional programs aimed to control antimicrobial use and reduce resistance [7,8]. The physician–patient relationship, clinical microbiology, health economics, and the most basic definitions of sickness and therapy are all important factors in antibiotic use [9].

Every healthcare member has particular responsibilities toward antibiotic stewardship [10]. Dentists write one in ten therapeutic antibiotic prescriptions in primary care, yet many of these prescriptions may be unneeded and exacerbate the critically important issue of bacterial resistance [11]. A study discovered that dentists’ awareness of recommendations and indications for prescribing antibiotics was often rated as moderate. These investigations revealed that even when they are not required or suitable, antibiotic prescriptions are common [12,13]. The nurses’ role is noted in several contexts, including in collaboration with other healthcare professionals, participation in prescribing choices, adherence to procedures, and patient and public education [14].

Future health workers should be educated about antimicrobial use and resistance during their undergraduate studies, and it is important to assess their knowledge and attitudes during their internship year [10]. They should have a well-developed awareness and adequate knowledge of antimicrobial resistance. Insufficient education on antimicrobial resistance may lead to serious medical problems for patients [10].

Many global studies have found areas of mistrust and knowledge gaps in prescribing antibiotics, and students recognize the need for additional learning and training in antimicrobial stewardship and prescribing [15]. Awareness and attitudes toward the correct usage of antibiotics between interns and other healthcare providers make a considerable difference in controlling antimicrobial resistance [16]. A previous study reported a high awareness of antimicrobial resistance among adult medical staff in Saudi Arabia [9]. According to previous studies, medical curricula focused on antibiotics’ rational use significantly improved students’ knowledge of antibiotics and discouraged inappropriate use of antibiotics in clinical practice [17]. In contrast, some studies concluded a low overall level of awareness of antibiotic use among residents in Saudi Arabia [18,19,20]. Many factors contribute to this level of awareness; among these factors is previous self-administration of antibiotics without a prescription. Additionally, lack of awareness is a major factor in the misuse of antibiotics worldwide [21].

Educating future practitioners on antibiotic use and antimicrobial stewardship is a global strategic goal. Promoting responsible antibiotic use is vital to limit the risk of antibiotic resistance [22,23]. However, it is critical to assess how effectively pharmacists, physicians, nurses, and dentists are educated and trained about antibiotics utilization in order to minimize the impacts of antimicrobial resistance on patient health. Currently, to the best of our knowledge, there is a lack of studies examining the perception of the above issues among pharmacy, medicine, nursing, and dental interns in Saudi Arabia. Therefore, this study aimed to evaluate the current pharmacy and non-pharmacy interns’ knowledge, attitudes, and practices regarding antimicrobial resistance and antibiotic use in Saudi Arabia.

## 2. Materials and Methods

### 2.1. Study Design and Setting

A cross-sectional descriptive study was conducted between January and April 2022 in Saudi Arabia. A self-administered questionnaire was used to assess the knowledge, attitudes, and practices of antibiotic use and antimicrobial resistance among pharmacy and non-pharmacy interns. The data were collected using the online survey tool Google Forms.

### 2.2. Sample Size

The sample size was estimated using the Raosoft Sample Size Calculator (http://www.raosoft.com/samplesize.html) (accessed on 1 January 2022) based on a margin of error of 5%, a confidence level of 90%, a population size consisting of the number of pharmacy and non-pharmacy interns in the kingdom, and a response distribution of 50%. The calculated sample size was 266. The total sample was proportionally allocated based on the number of healthcare graduate students in Saudi universities according to the Ministry of Health statistic report of approximately 15,000 [24].

### 2.3. Study Population

The inclusion criteria included pharmacy and non-pharmacy interns (medicine, dentistry, and nursing) who graduated from governmental or private Saudi universities.

The exclusion criteria included interns who had completed their internship year.

### 2.4. Data Collection Tool

Pharmacy and non-pharmacy interns received a soft copy of the survey questionnaire using the online survey platform Google Forms. After reviewing the literature [25,26,27] reporting on awareness and attitudes toward antibiotic use, we adopted the currently presented validated questionnaire because it has three different domains addressing the knowledge, attitudes, and practices of antibiotics. Each domain comprised professional questions inquiring about antibiotic consumption, perceptions, and general practice. The questionnaire was then reviewed by a faculty member expert in the field from the University of Tabuk to assess each domain’s component, content, clarity, relevance, and comprehensibility.

The questionnaire consisted of two sections. The first explored participants’ demographics, including gender, profession, college, and region. The second evaluated the interns’ knowledge, attitudes, and practices concerning antibiotic prescriptions and antimicrobial resistance. The second part had 3 domains; the knowledge domain was assessed using 15 items, the attitude domain was assessed using 8 items, and the third domain assessed practices on 5 items. All questions were close-ended, and participants’ responses were obtained for the knowledge and attitude domains on a 3-point response scale (agree, disagree, and not sure). The response to the practice domain was assessed through yes or no responses. The correct response to each question carried one mark. The overall score was graded as poor (≤50%), adequate (from 51 to <75%), or good ≥75% depending on the percentage of correct responses.

### 2.5. Ethical Considerations

The local Research Ethics Committee at the University of Tabuk officially approved this study (approval number, UT-186-41-2022). Informed consent was obtained from all participants.

### 2.6. Statistical Analysis

The internal consistency and reliability of the questionnaire were examined by calculating Cronbach’s alpha coefficient. The antibiotics questionnaire had a good internal consistency (Cronbach’s α = 0.81). The standard descriptive analysis was summarized to describe the demographic characteristics of the participants. A bivariate analysis using chi-square or Fisher exact test, when appropriate, was used to generate *p* values of the percentages of the correct responses for each item of the questionnaire between pharmacy and non-pharmacy interns. A *p* value of less than 0.05 was considered statistically significant. The sum scores of the correct responses per participant on each of the knowledge, attitude, and practice domains of antibiotic use were calculated. Then, the mean proportions and standard deviations of the three scores were calculated and tested for pharmacy and non-pharmacy interns using the Wilcoxon–Mann–Whitney test. The Spearman’s rank correlation was used to measure the direction of association between interns’ knowledge and their practices of antibiotic use.

All analyses were carried out using STATA SE V.11 (Stata Corp, College Station, TX, USA).

## 3. Results

### 3.1. Interns’ Demographic Characteristics and the Overall Average Score

A total of 266 respondents agreed to participate in this study, of whom 140 (52.6%) were male, and 244 (91.7%) attended governmental universities. The majority of participants were pharmacy interns (124, 46.6%), while 142 participants (53.4%) were interns at non-pharmacy schools, including schools in the medicine (87; 32.7%), dentistry (40; 15%), and nursing (15; 5.7%) domains. Many interns were from the middle region (76 interns; 28.5%) and the south region (62 interns; 23.3%). The characteristics of the study participants are illustrated in Table 1.

The average percentages of correct scores for pharmacy interns compared with non-pharmacy interns on the knowledge, attitude, and practice domains of the antibiotic use questionnaire are illustrated in (Table 2). The distribution of participants’ average scores in each domain of the questionnaire is shown in (Table 2).

On average, interns demonstrated good scores (≥75% correct response) on the knowledge and practice domains of the questionnaire and adequate scores (51–75% correct response) on attitude toward antibiotic use. The analysis showed that pharmacy interns demonstrated higher but insignificant differences in the percentage of correct scores on the knowledge, attitude, and practice domains of antibiotic use of the questionnaire. The percentage of correct responses for each item of the questionnaire is illustrated in Table 3, Table 4 and Table 5.

### 3.2. Knowledge of Antibiotic Use and Antibiotic Resistance

The pattern of average scores on each item of the knowledge domain showed that pharmacy interns achieved a higher percentage of correct responses than non-pharmacy medical interns. The scores of pharmacy interns were significantly higher for some questions inquiring about the association between the development of antibiotic resistance and appropriate antibiotic selection (*p* = 0.02) and the spread of nosocomial infection (*p* = 0.03). Pharmacy interns were more likely to be aware of the contribution of overconsumption of antibiotics to antibiotic resistance (*p* = 0.05). Two-thirds of interns (183; 68.8%) reported poor knowledge and confusion about whether antibiotic resistance can spread from animals to humans in question 15 without significant differences between professions.

### 3.3. Attitude of Interns toward Antibiotic Use and Antibiotic Resistance

A similar pattern of average correct responses was observed for questions regarding the attitude of interns toward antimicrobial prescription. The majority of responses reported a highly positive attitude among all professions. A negative attitude was observed toward the sole role of medical experts in solving antibiotic resistance issues. The average attitude score for all interns was adequate at 61.5%.

### 3.4. Practices of Interns Regarding Antibiotic Use and Antibiotic Resistance

The average practice score showed a good response (85.3%). Self-reported practices were appropriate, and there was no significant difference between pharmacy and non-pharmacy interns (Table 5).

The weakly positive and significant correlation between the percentage of correct scores on the practices of participants regarding antibiotic use and their score on knowledge of the use of antibiotics is illustrated in Figure 1 (Spearman’s rho = 0.32; *p* = 0.001).

## 4. Discussion

Globally, educating future healthcare professionals about antimicrobial use and resistance is a strategic objective [23]. With the increased occurrence of antibiotic prescriptions during the COVID-19 era, encouraging the responsible use of antibiotics is crucial to reduce the risk of antimicrobial resistance, and healthcare professionals are involved in the prescription, dispensing, administration, and use of antibiotics [22,28]. To reduce the negative effects of antimicrobial resistance on patients’ health, it is crucial to evaluate how well future pharmacists, physicians, nurses, and dentists are educated and trained on the proper use of antibiotics.

This cross-sectional study assessed the awareness, attitudes, and practices of pharmacy and non-pharmacy interns, including medicine, dentistry, and nursing interns, regarding antibiotic use and antibiotic resistance in Saudi Arabia. To the best of our knowledge, this is the first study in Saudi Arabia to assess interns’ knowledge of antibiotic use and antibiotic resistance. We chose interns as the participants in our study as they are future healthcare practitioners who will prescribe, dispense, monitor, and administer antibiotics. Assessing their awareness levels will help improve practices and minimize antibiotic misuse and resistance.

Our results reveal overall good levels of knowledge (76.1%) and practice (84.6%) and adequately positive attitudes (61.5%) among interns in Saudi Arabia, which reflect a satisfactory level of material in undergraduate curricula about antibiotic use and mechanism and causes of antimicrobial resistance.

In the current study, pharmacy interns had higher but insignificant scores in the knowledge domain than non-pharmacy interns. The scores of pharmacy interns were significantly higher for some questions inquiring about appropriate antibiotic selection, the spread of nosocomial infection, and the development of antibiotic resistance. A cross-sectional study was conducted to assess the infectious disease contents in pharmacy curricula at Saudi universities, including material on antibiotic use and antimicrobial stewardship [29]. They found that crucial infectious disease topics were covered in the curriculum by 78% of Saudi pharmacy colleges. These topics should prepare and enable graduates from these programs to enact better practices and provide better patient-centered care. This may also imply that pharmacists are distinctively qualified medical professionals who can play essential roles in the fight against antimicrobial misuse and resistance if they are involved in regulatory policies and educational interventions.

In this study, although all interns scored and expressed a good level of knowledge about antibiotic use, poor to adequate responses were reported for three questions regarding sending clinical samples before starting antibiotics, the efficacy of new antibiotics, and the spread of nosocomial infection. Unfortunately, our study showed that two-thirds of interns had poor knowledge and confusion about whether antibiotic resistance can spread from animals to humans. Similar results were reported by a previous study conducted in India, including medical interns [25]. This poor knowledge about the spread of antibiotic resistance from animals to humans demonstrates the need to promote awareness of this phenomenon in undergraduate curricula. At this stage, knowledge of antibiotics should be promoted through training, clinical rotations, or awareness workshops and activities. However, the knowledge of antibiotic use among medical students from different countries reported in the literature demonstrated variable results. Studies conducted in Colombia [26] and the United Arab of Emirates (UAE) [30] reported lower scores of knowledge among medical students, while a study conducted in Jordan reported a higher score (>75%) for students’ knowledge of antibiotic use and resistance [31]. This variation in the knowledge scores observed between different studies may be attributed to differences in the presence of materials about antimicrobials in curricula among universities and the recruited students’ performance and sociodemographic characteristics. However, the validated questionnaires used in these studies assessed the knowledge on antibiotic use and resistance; the minimal differences in the content may be a reason for the differences in the knowledge between the studies [26,30,31].

In the current study, despite the good knowledge of the participating interns, they had adequate scores for attitudes toward antibiotic use (61.5%). Moreover, some responses showed negative attitudes; half of the respondents believed that antibiotic resistance can be reduced by using broader-spectrum antibiotics and that they do not have a pivotal role in controlling the resistance, and 85% of the participants agreed that medical experts will solve the problem of antibiotic resistance before it becomes too serious. Adequate responses (51–75%) were scored for items related to the use of antibiotics and irrational practice. Similar attitudes were reported from studies conducted in India on medical students and interns [25] and medical students in Columbia [26]. However, a study conducted in Jordan on medical, pharmacy, and nursing students reported positive attitudes and that 62.8% of the participants believed that they have a role in controlling antibiotic resistance [31], and another study showed highly positive attitudes with a score of 80% among medical students in the United Arab Emirates (UAE) [30]. Pharmacy and medical schools in Saudi Arabia should offer more support to their undergraduate students by delivering more appropriate, well-designed curricula that address the proper use of antibiotics and the global risk of antimicrobial resistance. This will help promote the confidence of professional healthcare providers in dealing with antibiotics.

Regarding self-reported practices in this study, all interns expressed good levels with an average score of more than 80%, and there was no significant difference between pharmacy and non-pharmacy interns. However, our study showed that there was a positive correlation between the knowledge of interns and their practices of antibiotic use. This reflects the pharmacy and medical interns’ readiness to implement their awareness and beliefs in practice. On the other hand, a cross-sectional study conducted in Spain to identify factors that affect antibiotic prescribing practices among medical interns demonstrated the judgment of the attending physician was the main factor affecting medical interns’ antibiotic prescribing practices in this study [32].

Our study addressed the awareness and attitudes of pharmacy and non-pharmacy interns who shared many demographic characteristics, such as age, Saudi nationality, and socioeconomic status. The majority (92%) were enrolled in governmental universities and had limited experience in dealing with antibiotics, with the exception of their advanced training year. The demographic variable that was hypothesized to influence their awareness and attitudes toward antibiotic use was the undergraduate educational program (profession). Therefore, the study focused on this variable in the analysis. Moreover, the findings of the univariate analysis revealed no significant differences in the average scores of each domain, so a multivariate analysis was not necessary.

This study has the strength of being the first study to assess interns’ knowledge, attitudes, and practices regarding antibiotic use and resistance in Saudi Arabia. These interns are in their initial steps of professional practice. However, the current study has some limitations that should be acknowledged, mainly related to the study design. A cross-sectional survey design limited our ability to identify factors affecting the awareness and beliefs of interns. The small sample size of participants from different universities is another limitation. Third, the self-report scale used for data collection may lead to recall bias. The participants might overestimate some socially desirable attitudes or practices regarding antibiotic use.

## 5. Conclusions

The current study demonstrates that the knowledge, attitudes, and practices regarding antibiotic use and resistance among pharmacy and non-pharmacy interns in Saudi Arabia are adequate overall. In addition, knowledge was positively correlated with interns’ experience in practices related to antibiotic use. However, interns’ attitude toward their role and responsibility regarding the spread of antibiotic resistance is lacking. Further studies are needed to identify areas and effective means of development for pharmacy and non-pharmacy interns in Saudi Arabia.

## Figures and Tables

**Figure 1 healthcare-11-01283-f001:**
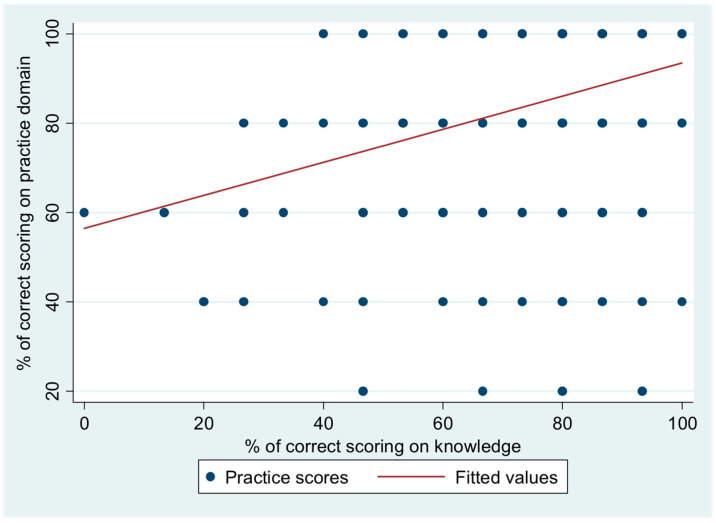
Correlation between percentage of correct scores on practice domain based on scores on knowledge.

**Table 1 healthcare-11-01283-t001:** Respondents’ demographic characteristics (*n* = 266).

Variables		Frequency	Percentage (%)
Gender	Male	140	52.6
	Female	126	47.4
Profession	Pharmacy	124	46.6
	Medicine	87	32.7
	Dentistry	40	15
	Nursing	15	5.7
Region	North	51	19.1
	East	35	13.1
	West	43	16.1
	Middle	76	28.5
	South	62	23.3
Type of University	Governmental	244	91.7
	Private	22	8.3

**Table 2 healthcare-11-01283-t002:** Pharmacy and non-pharmacy interns’ average correct scores on the knowledge, attitude, and practice domains of antibiotic use.

Domain	Total *n* = 266	Pharmacy *n* = 124	Non-Pharmacy *n* = 142	*p* Value *
	Mean score % (SD)			
Knowledge	76.1 (17.1)	78.3 (14.8)	74.2 (18.6)	0.13
Attitude	61.5 (23.2)	63.7 (23.1)	59.5 (23.3)	0.11
Practice	84.6 (20.5)	84.8 (21.4)	84.6 (19.8)	0.55

***** Significant *p* value < 0.05.

**Table 3 healthcare-11-01283-t003:** Frequency and percentage of correct responses for knowledge items.

Knowledge Items (Correct Response)	Pharmacy *n* = 124 (%)	Non-Pharmacy *n* = 142 (%)	Total *n* = 266 (%)	*p* Value *
1. Antibiotics are used to treat viral infections (D)	105 (84.7)	130 (91.6)	235 (88.4)	0.08
2. Antibiotics are useful against all types of common cold (D)	108 (87.1)	126 (88.7)	234 (87.7)	0.68
3. Clinical samples should be sent to culture and sensitivity before starting antibiotics (A)	86 (69.4)	94 (66.2)	180 (67.7)	0.58
4. The efficacy is better if the antibiotics are newer and more costly (D)	92 (74.2)	93 (65.5)	185 (69.5)	0.12
5. Antibiotics cause negative effects on the body’s own bacterial flora (A)	97 (78.2)	114 (80.3)	211 (79.3)	0.68
6. Incomplete antibiotic intake contributes to antibiotic resistance (A)	115 (92.7)	124 (87.3)	239 (89.9)	0.15
7. Inaccurate antibiotic selection contributes to antibiotic resistance (A)	112 (91.1)	115 (80.9)	277 (85.7)	0.02 *
8. Inadequate dosage of antibiotics contributes to antibiotic resistance (A)	107 (86.3)	114 (80.3)	221 (83.1)	0.19
9. Overprescription of antibiotics contributes to antibiotic resistance (A)	114 (91.9)	112 (85.9)	236 (88.7)	0.12
10. Overconsumption of antibiotics contributes to antibiotic resistance (A)	106 (85.5)	108 (76.1)	214 (80.5)	0.05
11. Nosocomial infection spread contributes to antibiotic resistance (A)	73 (58.9)	64 (44.1)	137 (51.5)	0.03 *
12. Self-medication of antibiotics contributes to antibiotic resistance (A)	107 (86.3)	113 (79.6)	220 (82.7)	0.15
13. Frequent use of same antibiotic will reduce the efficacy of the treatment (A)	78 (62.9)	98 (69.0)	176 (66.2)	0.29
14. Unnecessary use of antibiotics can lead to antibiotic resistance (A)	114 (91.9)	125 (88.0)	239 (89.9)	0.29
15. Antibiotic resistance can spread from animals to humans (A)	43 (34.9)	40 (28.2)	83 (31.2)	0.25

Note: (A) Agree, (D) Disagree. ***** Significant *p* value < 0.05.

**Table 4 healthcare-11-01283-t004:** Frequency and percentages of positive attitudes toward antimicrobial prescribing and resistance.

Attitude Items (Correct Response)	Pharmacy *n* = 124 (%)	Non-Pharmacy *n* =142 (%)	Total *n* = 266 (%)	*p* Value *
16. Antibiotics are safe drugs and, hence, can be commonly used (D)	94 (75.8)	92 (64.8)	186 (69.9)	0.051
17. Irrational antibiotic practice locally will not matter for global resistance (D)	88 (71.0)	104 (73.2)	192 (72.1)	0.68
18. Skipping one or two doses does not contribute to antibiotic resistance (D)	83 (66.9)	87 (61.2)	170 (63.9)	0.34
19. Antibiotic resistance can be reduced using higher antibiotics in spite of lower antibiotics being sensitive (D)	71 (57.3)	64 (45.1)	135 (50.8)	0.047 *
20. Medical experts will solve the problem of antibiotic resistance before it becomes too serious (D)	21 (16.9)	19 (13.4)	40 (15.1)	0.42
21. There is not much people, like me, can do to stop antibiotic resistance (D)	56 (45.2)	72 (50.7)	128 (48.1)	0.37
22. Adhering to antibiotic policy of the hospital will reduce the development of antibiotic resistance (A)	108 (87.1)	123 (86.6)	231 (86.8)	0.91
23. Antibiotic resistance is one of the biggest problems the world faces (A)	111 (89.5)	115 (80.9)	226 (85.0)	0.052

Note: (A) Agree, (D) Disagree. ***** Significant *p* value < 0.05.

**Table 5 healthcare-11-01283-t005:** Participant’s self-reported practices regarding antibiotic use.

Practice Items (Correct Response)	Pharmacy *n* = 124 (%)	Non-Pharmacy *n* = 142 (%)	Total *n* = 266 (%)	*p* Value *
24. Do you consult a doctor before starting an antibiotic? (Yes)	117 (94.4)	124 (87.3)	241 (90.6)	0.05
25. Do you prescribe the same antibiotic for relatives/friends for similar illnesses without consulting the doctor? (No)	105 (84.7)	120 (84.5)	225 (84.6)	0.97
26. Do you pressurize the doctor to prescribe antibiotics? (No)	98 (79.0)	122 (85.9)	220 (85.9)	0.14
27. Do you complete the full course of antibiotics treatment? (Yes)	109 (87.9)	118 (83.1)	227 (85.3)	0.27
28. Do you save the remaining antibiotics for the next time you get sick? (No)	97 (78.2)	116 (81.7)	213 (80.1)	0.48

* Significant *p* value < 0.05.

## Data Availability

Data are available upon request from the corresponding author.

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
