# Peer review of "Knowledge, Attitudes, and Practices among Pharmacy and Non-Pharmacy Interns in Saudi Arabia Regarding Antibiotic Use and Antibiotic Resistance: A Cross-Sectional Descriptive Study"

_healthcare, 2023, doi:10.3390/healthcare11091283_

Round 1
Reviewer 1 Report
I would thank the authors for this interesting study. However, the manuscript has shown some gaps that really bothered me.
One of the main limitation of the manuscript is the used questionnaire. In fact, the authors used a "basic" questionnaire updated from an article published in a non indexed journal with "simple" questions about antibiotic resistance.
The second concern is related to the statistical analysis. The authors used "simplistic analysis" to determine the association between the different scores and the demographic factors (limited to only 4 factors and only 1 factor (profession) was studied. Additionally, they should confirm their results is using multivariate analysis.
The third concern is related to the discussion which is also very simplistic. Also, using only 4 references for the discussion is insufficient especially in a such important field.
I have also other "minor comments":
Title: you can replace "Pharmacy, Dentistry, Medicine and Nursing Interns" by "medical interns".
The same remark applies to all the manuscript.
Also the authors tend to use the expression : "Pharmacy and other medical interns" in all the manuscript. Yet, they can simplify by using "medical interns".
Table 1: Delete the % from the first lines (gender and profession) and add it to the head of the table.
Figure 1 is not necessary. I suggest to delete it.
Table 2: You should indicate which is the significant result.
Discussion: (see above)
Line 226-229: " This poor knowledge……and activities". Are you discussion the general knowledge or only the knowledge about this question (confusion about whether antibiotic resistance can spread 224 from animals to humans) here?!!!
Line 234-237: These difference could be essentially due to the used items. Did they used the same questions? If not can we really compare?
Line 267- 268: "we assume…resistance". Delete this sentence please. We cannot assume so with a such number.
Reformulate the conclusion please.
Author Response
Manuscript ID healthcare-2331252
Point-by-point responses to the reviewer's comments
We would like to thank the editor and reviewer for the constructive critique and positive review of the manuscript. We sincerely appreciate all valuable comments and suggestions, which help us to improve the quality of the manuscript. We have considered all major and minor comments. The manuscript has been revised according to the suggested recommendations.
Below we have addressed the comments from reviewer #1. Our authors' responses are structured as follows: 1. Reviewer comment, 2. Author response, 3. Please see yellow highlighted text in the revised manuscript with relevant page and line numbers.
Please see revised manuscript in the attachment.
Reviewers’ #1 comments:
I would thank the authors for this interesting study. However, the manuscript has shown some gaps that really bothered me.
Comment 1: One of the main limitations of the manuscript is the used questionnaire. In fact, the authors used a "basic" questionnaire updated from an article published in a non-indexed journal with "simple" questions about antibiotic resistance.
Response:
References for relevant literature for developing questionnaires from indexed journals were added as a source for this study's questionnaire. Details in reviewing the questionnaire was added. Please see (Page 3; lines 111-118).
Comment 2: The second concern is related to the statistical analysis. The authors used "simplistic analysis" to determine the association between the different scores and the demographic factors (limited to only 4 factors and only 1 factor (profession) was studied. Additionally, they should confirm their results is using multivariate analysis.
Response:
Detailed explanation on demographic factors in this study was added in the discussion part. Please see (Page 10; line 287-295)
Comment 3: The third concern is related to the discussion which is also very simplistic. Also, using only 4 references for the discussion is insufficient especially in a such important field.
Response:
This study is intended to address the basics of antibiotics use and resistance in pharmacy and non-pharmacy trainees across the kingdom. Further evaluations will follow including more detailed variables and sophisticated statistics. The discussion part was enriched with new references that reflect different countries to target international readers. Please see discussion part (pages 8-10; lines 207-304)
Comment 4: I have also other "minor comments":
Title: you can replace "Pharmacy, Dentistry, Medicine and Nursing Interns" by "medical interns".
The same remark applies to all the manuscript.
Response:
The title was modified to "pharmacy and non-pharmacy interns” and made consistent with results and analysis. (Page 1, line 2)
Comment 5: Also, the authors tend to use the expression: "Pharmacy and other medical interns" in all the manuscript. Yet, they can simplify by using "medical interns".
Response:
"Pharmacy and other medical interns" were changed to pharmacy and non-pharmacy interns" to be consistent with results and analysis all over the manuscript. Please see (Page 1; line 19), (page 2; line 87), (page 3; lines 94, 100, 106, 110, 140), (page 4; lines 144, 153, 159), (page 5; line 174), (page 7; line 194), (page 8; lines 217, 229), (page 9; line 280). Content in the tables were changed accordingly.
Comment 6: Table 1: Delete the % from the first lines (gender and profession) and add it to the head of the table.
Response:
Percentage (%) was deleted and added to the head of the table (Page 4; Table 1)
Comment 7: Figure 1 is not necessary. I suggest to delete it.
Response:
Figure 1 was deleted.
Comment 8: Table 2: You should indicate which is the significant result.
Response:
Significant results were bolded and superscripted with star; footnote was added below each table.
Comment 9: Line 226-229: " This poor knowledge……and activities". Are you discussion the general knowledge or only the knowledge about this question (confusion about whether antibiotic resistance can spread 224 from animals to humans) here?!!!
Response:
The poor knowledge reflect question #15 (antibiotic resistance can spread from animals to humans). The sentence was edited to further clarify this point in the discussion part. (Page 9; line 246)
Comment 10: Line 234-237: These difference could be essentially due to the used items. Did they used the same questions? If not can we really compare?
Response:
Detailed explanation on the reasons of the observed difference was added in the discussion part. (Page 9; line 257)
Comment 11: Line 267- 268: "we assume…resistance". Delete this sentence please. We cannot assume so with a such number.
Response:
The sentence was deleted.
Comment 12: Reformulate the conclusion please.
Response:
The conclusion was reformulated in consistence with the study findings. (Page 1; lines 305-312)

Reviewer 2 Report
I’d suggest to change title because it is bit confusing. It should be "pharmacy vs no pharmacy interns" as per results and analysis.
Authors are requested to collect and add more information from the papers/literature that are published in 2020-2023. The article lacks in this area.
The objectives for the study and the rationale for the work and the research gap being addressed need to be better articulated.
Results are also not clearly presented as per study title.
I'd suggest to remove both figures. Fig 1 is useless and Fig 2 is not depicting the excet information
Please revise your document to provide more consistent arguments according to new title. At times, focus on information not directly linked to the aims of your article or the value of your data.
Globally I suggest you write your article towards the international readership of your targeted journal.
I also encourage the authors to have their manuscript thoroughly proofread by a native English speaker prior to resubmission as there are lot of mistakes.
Author Response
Manuscript ID healthcare-2331252
Point-by-point responses to the reviewer's comments
We would like to thank the editor and reviewer for the constructive critique and positive review of the manuscript. We sincerely appreciate all valuable comments and suggestions, which help us to improve the quality of the manuscript. We have considered all major and minor comments. The manuscript has been revised according to the suggested recommendations.
Below we have addressed the comments from reviewer #2. Our authors' responses are structured as follows: 1. Reviewer comment, 2. Author response, 3. Please see yellow highlighted text in the revised manuscript with relevant page and line numbers.
Please see revised manuscript in the attachment.
Reviewers’ #2 comments:
Comment 1: I’d suggest to change title because it is bit confusing. It should be "pharmacy vs no pharmacy interns" as per results and analysis.
Response:
The title was modified to "pharmacy and non-pharmacy interns". (Page 1; line 2)
Comment 2: Authors are requested to collect and add more information from the papers/literature that are published in 2020-2023. The article lacks in this area.
Response:
Discussion was enriched with recent literature as cited in references (Perrella A, Fortinguerra 2023, Cantón, R.; Akova 2022, Molina-Romera 2023). Please see the discussion section (Pages 8-10; lines 207-304)
Comment 3: The objectives for the study and the rationale for the work and the research gap being addressed need to be better articulated.
Response:
Point addressed. Please see introduction (page 2; lines 80-84)
Comment 4: Results are also not clearly presented as per study title.
Response:
Title was modified and results are now better aligned with the title.
Comment 5: I'd suggest to remove both figures. Fig 1 is useless and Fig 2 is not depicting the exact information
Response:
Figure 1 has been deleted. However, we prefer to keep Fig 2 as it describes the direction of association between interns’ knowledge and their practice of antibiotic use. Detailed statistical analysis was added to the methods section. Figure 2 changed to Figure 1 now (page 8; line 204) in leu of this edit.
Comment 6: Please revise your document to provide more consistent arguments according to new title. At times, focus on information not directly linked to the aims of your article or the value of your data.
Response:
The document has been revised and thoroughly checked for alignment with the edited title as well as the accuracy of arguments based on the aims and objectives of this protocol as well as the objective findings and data.
Comment 7: Globally I suggest you write your article towards the international readership of your targeted journal.
Response:
The discussion part was enriched with new references that reflect different international practices to target international readers. Please see discussion part (pages 8-10; lines 207-304).
Comment 8: I also encourage the authors to have their manuscript thoroughly proofread by a native English speaker prior to resubmission as there are lot of mistakes.
Response:
The manuscript has undergone English language editing by MDPI. The text has been checked for correct use of grammar and common technical terms, and edited before resubmission.

Reviewer 3 Report
I'm happy to review this manuscript.
This work describes the difference of knowledge , attitude and practice between interns in pharmacy and other health professionals.
The topics meet with journal's scope.
Results would be reasonable but I have some concern for study design.
The authors calculated the sample size based on one ratio. However, they carried out hypothesis testing repeatedly in each item in the manuscript. Therefore the power of tests are likely to be significantly lower. The p-value of difference between knowledge of pharmacy interns and other medical interns (p=0.047) other significant tests should be discussed based on lower power of the test. Oveall the reviewer believes a different sample size calculation scenario is needed.
The sample size calculation was based on 90% CI revel but the results were used 95%CI. The authors should be consistent through the manuscript.
Figure 2 indicated average score of each domain in pharmacy interns and other medical interns. The authors used t-test however the reviewer are not sure that t-test was suitable for the analysis between two averages. At least, SD, not 95%CI, of average of each domain is necessary to show distribution of mean score.
In figure 3, the author showed correlation between % correct score of knowledge and practice. Please show the method (Peason, Spearman or others) and appropriateness to calculate correlation coefficient in the statistical section.
Line 255: the authors discussed that the average score of more than 80% was appropriate level. Please show why more than 80% was considered to be appropriate level.
Author Response
Manuscript ID healthcare-2331252
Point-by-point responses to the reviewer's comments
We would like to thank the editor and reviewer for the constructive critique and positive review of the manuscript. We sincerely appreciate all valuable comments and suggestions, which help us to improve the quality of the manuscript. We have considered all major and minor comments. The manuscript has been revised according to the suggested recommendations.
Below we have addressed the comments from reviewer #3. Our authors' responses are structured as follows: 1. Reviewer comment, 2. Author response, 3. Please see yellow highlighted text in the revised manuscript with relevant page and line numbers.
Please see revised manuscript in the attachment.
Reviewers’ #3 comments:
I'm happy to review this manuscript.
This work describes the difference of knowledge, attitude and practice between interns in pharmacy and other health professionals.
The topics meet with journal's scope.
Results would be reasonable but I have some concern for study design.
Comment 1: The authors calculated the sample size based on one ratio. However, they carried out hypothesis testing repeatedly in each item in the manuscript. Therefore, the power of tests are likely to be significantly lower. The p-value of difference between knowledge of pharmacy interns and other medical interns (p=0.047) other significant tests should be discussed based on lower power of the test.
Overall, the reviewer believes a different sample size calculation scenario is needed.
Response:
Thank you for your note. Yes, the calculated post-hoc power of the observed difference in knowledge scores between pharmacy and non-pharmacy interns was insufficient to report a significant difference. Moreover, because of the left skewed scores of interns towards high knowledge, we tested the difference using Wilcoxon-Mann-Whitney test and new p values were reported in Table 2. (Page 4; line 163)
Table 2: Pharmacy and non-pharmacy interns' average correct scores on knowledge, attitude, and practice domains of antibiotic use
|
Domain |
Total n= 266 |
Pharmacy interns n= 124 |
Non-pharmacy interns n=142 |
p value |
|
|
Mean score (SD) |
|
||
|
Knowledge |
76.1 (17.1) |
78.3 (14.8) |
74.2 (18.6) |
0.13 |
|
Attitude |
61.5 (23.2) |
63.7 (23.1) |
59.5 (23.3) |
0.11 |
|
Practice |
84.6 (20.5) |
84.8 (21.4) |
84.6 (19.8) |
0.55 |
Comment 2: The sample size calculation was based on 90% CI level but the results were used 95%CI. The authors should be consistent through the manuscript
Response:
Sample size calculation was roughly estimated to get basic understanding of sample needed in this type of studies. We reported the average score of each domain for pharmacy and non-pharmacy interns with standard deviations as shown in Table 2 (Page 4)
Comment 3: Figure 2 indicated average score of each domain in pharmacy interns and other medical interns. The authors used t-test however the reviewer are not sure that t-test was suitable for the analysis between two averages. At least, SD, not 95%CI, of average of each domain is necessary to show distribution of mean score.
Response:
Due to the left skewed scores of interns towards high knowledge, the average score of each domain between pharmacy interns and other medical interns were tested using Wilcoxon-Mann-Whitney test and new p values were illustrated in Table 2. Standard deviations were reported beside the mean scores. (Page2)
Comment 4: In figure 3, the author showed correlation between % correct score of knowledge and practice. Please show the method (Pearson, Spearman or others) and appropriateness to calculate correlation coefficient in the statistical section.
Response:
Spearman correlation coefficient was calculated to measure the direction of association between interns’ knowledge and their practice of antibiotic use and it was added to the Method section. Figure 2 changed to Figure 1 now (page 8; line 204).
Comment 5: Line 255: the authors discussed that the average score of more than 80% was appropriate level. Please show why more than 80% was considered to be appropriate level.
Response:
It was described in the Method section in lines 127-129 that the overall score was graded as poor if the percentage of correct responses (≤50%), adequate (from 51 to <75%), and good if ≥75% were correct responses. Therefore, the sentence has been rephrased to “all interns expressed good levels with an average score of more than 80%,” (page 9; line 278)

Round 2
Reviewer 1 Report
Thank you for adressing most of my comments.
1. I have just a question related to the comparison. It is not better to compare between each category (pharmacy, medicine, ...) alone? and thus the tiltle will be "....amon medical interns...).
2. please delete the scores from the abstract and let just the percetages (we do not know what is your scale when we read the abstract).
3. Table 5: add an asterix for 0.05 (first line)
4. You should improve the quality of the tables (try to make the items in one lines (max 2 lines) to reduce their size.
Reviewer 2 Report
I'd be happy to accept paper
Author Response
Thank you. We highly appreciated the quality comments offered by Reviewer 2 towards improving our manuscript.
Reviewer 3 Report
The review thinks that this manuscript can be acceptable for publication.
Author Response
Thank you. We highly appreciated the quality comments offered by reviewer 3 towards improving our manuscript.
The manuscript has undergone English language editing by MDPI. The text has been checked for correct use of grammar and common technical terms, and edited before resubmission.